# Transcriptomics of Improved Fruit Retention by Hexanal in ‘Honeycrisp’ Reveals Hormonal Crosstalk and Reduced Cell Wall Degradation in the Fruit Abscission Zone

**DOI:** 10.3390/ijms22168830

**Published:** 2021-08-17

**Authors:** Karthika Sriskantharajah, Walid El Kayal, Davoud Torkamaneh, Murali M. Ayyanath, Praveen K. Saxena, Alan J. Sullivan, Gopinadhan Paliyath, Jayasankar Subramanian

**Affiliations:** 1Department of Plant Agriculture, University of Guelph, 50 Stone Road E, Guelph, ON N1G2W1, Canada; sriskank@uoguelph.ca (K.S.); davoud.torkamaneh.1@ulaval.ca (D.T.); ayyanath@uoguelph.ca (M.M.A.); psaxena@uoguelph.ca (P.K.S.); asulliva@uoguelph.ca (A.J.S.); gpaliyat@uoguelph.ca (G.P.); 2Department of Plant Agriculture, University of Guelph-Vineland Station, 4890 Victoria Ave N, Vineland, ON L0R2E0, Canada; we21@aub.edu.lb; 3Faculty of Agricultural and Food Science, American University of Beirut, Riad El Solh, P.O. Box 11-0236, Beirut 1107 2020, Lebanon; 4Faculté des Sciences de l’Agriculture et de l’alimentation, Universite Laval, Pavillon Paul-Comtois, 2425, rue de l’Agriculture, Local 1122, Québec City, QC G1V 0A6, Canada

**Keywords:** abscisic acid, cell-wall hydrolases, ethylene, fruit abscission zone, hexanal, transcriptome

## Abstract

Apples (*Malus* *domestica* Borkh) are prone to preharvest fruit drop, which is more pronounced in ‘Honeycrisp’. Hexanal is known to improve fruit retention in several economically important crops. The effects of hexanal on the fruit retention of ‘Honeycrisp’ apples were assessed using physiological, biochemical, and transcriptomic approaches. Fruit retention and fruit firmness were significantly improved by hexanal, while sugars and fresh weight did not show a significant change in response to hexanal treatment. At commercial maturity, abscisic acid and melatonin levels were significantly lower in the treated fruit abscission zone (FAZ) compared to control. At this stage, a total of 726 differentially expressed genes (DEGs) were identified between treated and control FAZ. Functional classification of the DEGs showed that hexanal downregulated ethylene biosynthesis genes, such as S-adenosylmethionine synthase (*SAM2*) and 1-aminocyclopropane-1-carboxylic acid oxidases (*ACO3*, *ACO4*, and *ACO4-like*), while it upregulated the receptor genes *ETR2* and *ERS1*. Genes related to ABA biosynthesis (*FDPS* and *CLE25*) were also downregulated. On the contrary, key genes involved in gibberellic acid biosynthesis (*GA20OX-like* and *KO)* were upregulated. Further, hexanal downregulated the expression of genes related to cell wall degrading enzymes, such as polygalacturonase (*PG1*), glucanases (endo-β-1,4-glucanase), and expansins (*EXPA1-like*, *EXPA6*, *EXPA8*, *EXPA10-like*, *EXPA16-like*). Our findings reveal that hexanal reduced the sensitivity of FAZ cells to ethylene and ABA. Simultaneously, hexanal maintained the cell wall integrity of FAZ cells by regulating genes involved in cell wall modifications. Thus, delayed fruit abscission by hexanal is most likely achieved by minimizing ABA through an ethylene-dependent mechanism.

## 1. Introduction

Apple (*Malus domestica* Borkh.) is one of the most widely cultivated fruits and ranks third in global fruit production [1]. Most apple trees tend to shed fruits just before the harvest, often referred to as preharvest fruit drop (PFD), which renders a huge economic loss to growers. The PFD usually begins 3–4 weeks before the anticipated harvest and causes yield losses up to 30% at the beginning of the harvest [2,3]. The severity of the PFD is cultivar specific and influenced by several factors such as mineral nutrients, summer pruning, water availability, growing season temperatures [2], lower starch content, and higher internal ethylene concentration [4]. Moreover, this issue is exacerbated when the fruits are left on the tree for better colour development, as colour has a huge consumer appeal. ‘Honeycrisp’, a premium apple cultivar, is categorized as more prone to PFD [5], which causes yield losses of almost 50% in some years [6].

PFD is a consequence of abscission, whereby cell separation occurs rather pre-maturely at the constriction region of the pedicel, resulting in fruit drop [7,8]. The PFD control measures in apples have largely relied upon the use of plant growth regulators (PGRs) and are often cultivar specific. Moreover, abscission is an irreversible physiological process, thus warranting suitable technologies to improve fruit retention. The application of hexanal as an aqueous formulation at the preharvest stage has shown promising results in improving fruit retention in several fruits, including apple [9], raspberry [10], mango [11], and orange [12]. Hexanal application also extends the shelf life of several horticultural commodities through inhibiting the membrane degradation enzyme, phospholipase D [13,14]. Previous studies on fruit abscission have reported that the cell-separation within the FAZ is the result of a cascade of physiological events due to the coordinated expression of PGR related genes resulting in abrupt changes in endogenous levels of plant hormones [7,15,16].

Plant hormone ethylene is the key regulator of abscission [17,18]. The application of ethylene inhibitors such as aminoethoxyvinylglycine (AVG) and 1-methylcyclopropene (1-MCP) reduced the PFD in ‘Bisbee Delicious’ apples [19]. On the contrary, ethylene releasing compound, ethephon, promoted fruit abscission in ‘Golden Delicious’ [20] and ‘Honeycrisp’ [21] apples, indicating that both ethylene biosynthesis and signalling pathways are involved in fruit abscission. Ethylene production and fruit softening increased more rapidly during fruit ripening of PFD-prone cultivar ‘Golden Delicious’ than non-prone cultivar ‘Fuji’ [22]. Moreover, transcript levels of ethylene biosynthesis genes *MdACS5A* and *MdACO1* as well as receptor genes *MdETR2* and *MdERS2* increased in the FAZ of ‘Golden Delicious’ [22]. Similarly, early induction of fruit abscission in melon was associated with upregulated expression of *SAMS*, *ACS*, *ACO*, and *ETRs* in the FAZ [23]. However, efforts to suppress the expression of ethylene related genes using PGRs effectively worked only in combined applications [24,25,26]. Previous studies have observed that hexanal decreased ethylene production in the ripening fruits [9,27] and significantly downregulated the expression of the *ACS* gene [28]. However, in spite of its promise in improving fruit retention, there is no information on how hexanal regulates the fruit abscission.

Plant hormones, particularly abscisic acid (ABA), auxin, and gibberellins (GA), also play substantial roles in fruit abscission. ABA levels generally increase towards fruit maturity and ripening and contribute to senescence and seed dormancy [29]. Although a high level of ABA in the AZ, prior to abscission, has been reported in several species, the direct involvement of ABA in the abscission process remains unclear. It has been suggested that ABA acts as a modulator of ACC levels, and therefore stimulates ethylene biosynthesis, leading to increased abscission [30,31]. Exogenous application of ABA contributed to PFD in ‘Golden Delicious’ apples [32]. Abscission was delayed in the ethylene-JA-ABA deficient triple mutant in Arabidopsis and showed an association between abscission and ABA levels [33]. In addition to ethylene and ABA, other phytohormones including auxin [8,15,34] and jasmonic acid (JA) [33] contributed to improving fruit retention in various crops.

Cell wall breakdown and cell separation are required within the FAZ for the fruit to abscise from the tree. Cell wall hydrolysis enzymes, such as polygalacturonase (PG) and cellulase (EG) pectate lyase, and cell wall loosening enzymes such as expansins promote fruit detachment at the AZ [35,36]. Ethylene is strongly correlated with the activity of these hydrolases and the expression of genes, including *MdPG2* and *MdEG1*, in the FAZ [22]. These findings suggest that ethylene plays a regulatory role in fruit abscission and can accelerate the abscission process. Ethylene also interacts with other plant hormones in fruit drop, particularly with ABA [32] and auxin [34].

We previously reported that the preharvest hexanal spray improved postharvest storage attributes of ‘Honeycrisp’ apples [9]. Based on those studies, we hypothesize that hexanal improves fruit retention either by regulating the abscission through an ethylene dependent mechanism or directly manipulating the ABA/GA mechanism. To test this hypothesis, we studied the mechanism of action of hexanal in improving fruit retention in ‘Honeycrisp’ using physiological, biochemical, and transcriptomic characterization. The differentially expressed genes (DEGs) due to the application of hexanal in the FAZ were identified via RNA-seq analysis. The functional profiling of the DEGs were studied through gene ontology annotation and enrichment analysis. Plant hormones present in the FAZ were quantified using reverse-phase ultra-performance liquid chromatography-mass spectrometry.

## 2. Results

### 2.1. Effect of Hexanal on Fruit Retention and Fruit Quality

Hexanal application significantly and consistently increased fruit retention in ‘Honeycrisp’ (Figure 1). All trees showed a continuous decline in fruit retention throughout the study period, but the rate of decline was significantly slower in the treated trees than in control. During the first 14 days of sampling, there was a steep drop in control trees (24–29%), then the drop seemed to slow down, whereas the drop was low (7–9%) and more stable in the treated fruits. At the end of the fruit retention study period of 49 days, treated trees retained three quarters of the total fruits while control trees retained less than half of the total fruits. Under field conditions, it was very noticeable that control fruits were starting to show cracks and were much softer while the hexanal treated fruits did not show any of these defects.

The effect of hexanal on fruit quality and physiological parameters that determine the maturation and ripening of ‘Honeycrisp’ apples were analyzed and presented in Table 1. As expected, hexanal treated fruits had significantly higher firmness than control fruits. The firmness rapidly declined in the control fruits while the decline was very gradual in the hexanal treated fruits. However, other quality parameters, such as total soluble solids (TSS) and fresh weight, did not vary significantly between treated and control fruits at commercial maturity and 49 days after commercial maturity, indicating that hexanal does not alter any other fruit quality traits (Table 1).

### 2.2. Quantitation of Plant Hormones in Fruit and FAZ

Phytohormones present in the fruit and FAZ were quantified in order to better understand the hormonal regulations underlying the correlation between hexanal application and the improvement in fruit retention. For hormonal analysis, we used the samples that were harvested at commercial maturity as our intention here is to study the hormonal changes that occur in the fruit and FAZ at commercial maturity (Figure 2a), at which time it is anticipated that ethylene and ABA levels will be high, based on earlier observations [2,22]. Ethylene evolution rate was not statistically different (*p* = 0.0879) between hexanal treated and control fruits at commercial maturity stage (Table 2). At this stage, hexanal treatment significantly reduced both abscisic acid (ABA) and melatonin concentrations in the FAZ. However, hexanal did not alter the zeatin concentration in the FAZ (Table 2). We could not detect any other plant hormones, such as indole-3-acetic acid (IAA), gibberellic acid (GA3), salicylic acid (SA), and jasmonic acid (JA), in the FAZ samples. Presumably, those are present below the detection limit of the reverse-phase ultra-performance liquid chromatography (UPLC) conducted with an Aquity QDa single quadruple mass spectrometer (MS) controlled by Empower 3 (Waters limited, Mississauga, ON, Canada)

### 2.3. Identification of Differentially Expressed Genes

RNA-seq analysis was conducted to study the variations in hexanal regulated genes expressions and their functions that are primarily related to hormonal regulation and associated functional pathways at the commercial maturity stage (Figure 2a). Mapping the rRNA depleted 96.89 million RNA-seq reads from our samples against the apple reference genome (*Malus domestica*, GDDH13 v1.1.) showed that 92.99 million reads (96%) were mapped in total. The mean mapped reads per sample were 23.25 ± 0.28 million. After removing low expressed genes with less than ten raw reads across all samples, we identified 30,709 genes in at least one of the samples (Appendix A). EdgeR modelling following the ComBat batch correction and trimmed median of means (TMM) normalization yielded 726 DEGs between hexanal treated and control samples at *p* ≤ 0.05 with an expression foldchange cut off (FC) ≥ 2. Among the 726 DEGs, 353 were upregulated (*p* ≤ 0.05; |log_2_foldchange ≥ 1|) while 373 were downregulated (*p* ≤ 0.05; |log_2_foldchange ≥ −1|) (Figure 2b,c, Appendix A) by hexanal.

### 2.4. Identification of Enriched Gene Ontology (GO) and Functional Pathways

In order to understand the functions of the DEGs, gene ontology (GO) functional classes and pathway enrichment analyses were conducted. We used *Arabidopsis thaliana* ortholog genes of the *Malus domestica* DEGs for this analysis. A total of 659 *Arabidopsis thaliana* ortholog genes were retrieved from the 726 *Malus domestica* DEGs. Enriched functional pathways were identified under all three GO term functional classes, biological process (BP), cellular component (CC), and molecular function (MF). The highest number of enriched pathways, representing 68% of the total enriched GO pathways, were identified under BP (231 pathways), while the lowest was under CC (nine pathways; 3%) functional classes (Table S3). Interactive enrichment networks plot (Figure 3) and hierarchical clustering (Appendix A) of first 30 top pathways of BP at *p* < 0.001 (FDR) showed the relationship and correlation among the significant pathways, respectively. Although various pathways could be identified under each GO term class, we focused on enriched specific pathways that could contribute to fruit retention at commercial maturity. These pathways were grouped into three main categories: plant hormone responses, transcription factors, and cell-wall modification. The DEGs belonging to selected enriched functional pathways were further characterized.

### 2.5. Characterization of Genes Related to Various Plant Hormone Responses

A total of 61 DEGs (30 and 31 up and downregulated) related to various plant hormone responses were identified and grouped. Of these, genes related to ethylene (16), ABA (16), and auxin (nine) were the most represented, followed by JA (six), GA (five), SA (four genes), cytokinin (four), and BR (one) as a result of hexanal treatment (Appendix A). Four genes involved in the ethylene biosynthesis pathway were identified, and all were downregulated by hexanal, including S-adenosylmethionine synthase 2 (*SAM2*) and three 1-aminocyclopropane-1carboxylic acid oxidases (*ACO3*, *ACO4*, and *ACO4-like*). Two classes of ethylene receptors (*ETR2-like*, *ERS1*) were also identified, and both were up-regulated by hexanal. Ethylene signaling pathway elements were differentially expressed. Some transcriptional activators, e.g., *AP2*/*ERF023* and *MYB113-like*, were down-regulated, while others (*AP2*/*ERF017*, *AP2*/*ERF5*, *AP2*/*EREBP6*, *AP2*/*EREBP105* and *AP2/ERF-B3-RAV1*) were upregulated by hexanal (Figure 4a).

Out of 16 ABA-related DEGs, hexanal up and downregulated four and twelve genes, respectively (Figure 4b). However, two key ABA biosynthetic genes, farnesyl diphosphate synthase (*FDPS*) and CLAVATA3-related protein-25 (*CLE25*), were significantly downregulated by three and four folds, respectively. Hexanal also downregulated ten genes related to ABA signalling, including serine/threonine phosphatases PP2C (*PP2C-34-like*, *PP2C-73*), ninja-family protein-*AFP1-like*, and ABC transporter G family member, *ABCG12-like*. On the contrary, most of the GA related genes were upregulated. The key genes involved in GA biosynthesis, such as gibberellin 20 oxidase1-like (*GA20OX-like*) and ent-kaurene oxidase (*KO*) were upregulated by nine and two folds, respectively by hexanal. Besides, a receptor gene *GIDIB* was upregulated by two folds compared to control (Figure 4d). Gene expression showed a divergent pattern in auxin biosynthesis, signalling, transport, and response. Genes related to auxin biosynthesis, tryptophan aminotransferase-related protein 3 (*TAR3*), and response (*CHS*, *MYB113-like*, and *CYP75B1*) were downregulated by hexanal in the FAZ, while genes involved in auxin transports, such as protein big grain 1-like (*BG1*), protein walls are thin-1 (*WAT1*), and thermospermine synthase-5, (*ACL5*) were upregulated (Figure 4c).

Altogether, fifteen DEGs were identified related to SA, JA, cytokinin, BR biosynthesis and signalling. The expression of these genes was greatly varied. For example, three out of four genes related to SA were up regulated, whereas three out of six genes related to JA were downregulated with the expression fold change between two and seven. Genes related to cytokinin were mostly downregulated (Appendix A). Since none of these genes have been associated with fruit abscission, they were not studied further. The above results suggested eight classes of plant hormones were altered by hexanal at the commercial maturity stage.

### 2.6. Characterization of Genes Encoding for Transcription Factors

Genes encoding transcription factors (TFs) were also studied to identify their response to the hexanal treatment. A total of 21 genes putatively encoding TFs of diverse families were differentially expressed (17 and four up and downregulated, respectively) in the FAZ (Appendix A). Of those, most genes (eight) belonged to the APETALA2/ERF (AP2/ERF) superfamily. Seven genes representing the AP2/ERF family were upregulated while one was downregulated by hexanal. These genes either act as repressors (*AP2/ERF4*) or activators (*AP2/ERF017*, *AP2/EREBP6*, *AP2/ERF5*, *AP2/ERF023*, *AP2/EREBP105 and AP2/ERF-B3-RAV1*) of GCC-box mediated gene expression in the ethylene-activated signalling pathway. Besides, a few genes that belonged to TF families bHLH, GATA, MADS-box, MYB, TCP, and WRKY were also identified. *MYB113-like* gene representing MYB transcription factor family, participating in the anthocyanin biosynthetic process, was downregulated by hexanal at harvest.

### 2.7. Characterization of Genes Related to Cell Wall Modification

DEGs that are related to cell wall modifications were also characterized to understand their functions in hexanal regulated cell wall integrity of the FAZ cells. A total of 31 genes were differentially expressed (six and 25 up and downregulated, respectively) at commercial maturity (Figure 5, Appendix A). The genes encoding enzymes related to callose, polygalacturonase, and expansins were downregulated by hexanal. Of those downregulated genes, eight were related to callose degradation, including endoglucanase 19-like, endo 1,4-β-glucanase, endo-glucanase-45-like, and glucan endo-1,3 β-glucosidase 8-like. Two genes were related to polygalacturonase (PG), including polygalacturonase 1 and endo-polygalacturonase-like-protein-like. Further, seven genes were related to expansin, including *EXPA1-like*, *EXPA6*, *EXPA8*, *EXPA10-like*, *EXPA16-like* (Figure 5). All these genes encoding enzymes related to callose, PGs, and expansins could be involved in maintaining the cell wall integrity of the FAZ cells.

### 2.8. Characterization Genes Related to Abscission

Two genes specifically involved in abscission were identified. Senescence-associated carboxylesterase 101-like (*SAG101-like*; MD17G1039700) encodes an acyl hydrolase involved in senescence/floral organ abscission that was downregulated. In contrast, zinc finger protein 2-like (*ZFP2-like*; MD00G1113500), that acts as a negative regulator of floral organ abscission, was upregulated by hexanal.

### 2.9. Confirmation of Gene Expression Patterns by qRT-PCR

To further confirm the RNA-seq data, selected genes representing ethylene biosynthesis and signalling pathway and cell wall metabolism were quantified using qPCR in the FAZ samples. All tested genes belonging to ethylene biosynthesis and signalling (*SAM2;* MD13G1141700, *ACO3;* MD09G1114800, *ETR2-like*; MD13G1209700 and *AP2*/*ERF17;* MD15G1221100) (Figure 6a–d) and cell wall metabolisms (*EXPA6*; MD03G1090700, *EXPA8*; MD07G1233100, *EG19-like;* MD06G1105900 and *1*,*4-β-EG3;* MD10G1003400) (Figure 6e–h) were confirmed by the qRT-PCR analysis. The results further revealed the decreased expression of six genes related to ethylene biosynthesis (*SAM2*, *ACO3*) and cell wall modification (*EXPA6*, *EXPA8*, *EG19-like*, and *1*,*4-β-EG3).* In contrast, ethylene receptor gene *ETR2-like* and ethylene responsive factor *AP2*/*ERF017* showed increased expression by hexanal. These data confirmed the presence of the above tested genes in the FAZ, and their expression patterns confirm the RNA-Seq results.

## 3. Discussion

### 3.1. Delay of Fruit Ripening and Ethylene Production by Hexanal

Fruit dropping shortly before harvest is a challenge to apple growers due to significant economic losses. In climacteric fruits like apples, the production of ethylene by the ripening fruits stimulates the production of cell wall degrading enzymes and forms an abscission zone in the pedicel [8]. Growers extensively resort to multiple PGR applications to reduce ethylene production, slow down the ripening, and improve fruit retention. However, the application of PGRs has its own limitations and further increases the cost of production [22,26]. ‘Honeycrisp’ is a premium apple variety that fetches higher returns for the growers but is also more prone to preharvest fruit drop. Hence, reducing the preharvest drop in such premium varieties will boost the economic returns for the growers as well as reduce preventable food loss.

Hexanal treated fruits retained firmness for a longer time compared to control fruits (Table 1). Moreover, hexanal spray had a striking effect on fruit retention for an extended time (Figure 1). Extending fruit retention time would be valuable to growers as it can extend the harvesting window and reduce postharvest loss. Moreover, enhanced fruit firmness is an added advantage for premium apple varieties like ‘Honeycrisp’ as they are mainly cultivated for the fresh market. However, although hexanal significantly reduced the ethylene evolution during storage of other climacteric fruits such as mango [27] and banana [38], we could not detect a significant treatment effect at commercial maturity. The improvement in fruit retention and firmness due to hexanal may be associated with a slowdown in the ripening process, thus delaying the abscission. Further, ‘Honeycrisp’ is not amenable to controlled atmospheric storage (CA) [9,22], and thus it warrants alternate methods of extending shelf life, without developing bitter pit.

### 3.2. Hexanal Reduces Ethylene Biosynthesis and Perception in the FAZ

Ethylene biosynthesis increases before abscission in many senescing organs, including fruits [35]. In apple fruitlet, the application of chemical thinner ethephon stimulated the ethylene biosynthesis in parallel with the upregulation of key regulatory genes, *MdACO1*, *MdACS5A*, and *MdACS5B* in the FAZ [39], suggesting that ethylene biosynthesis and signaling in FAZ is involved in abscission. We have identified four key regulatory genes involved in the ethylene biosynthesis pathway, including *SAM2*, *ACO3*, *ACO4*, and *ACO4-like* (Figure 4a, Appendix A). Interestingly, the transcript levels of all four genes were substantially decreased by hexanal at commercial maturity. A similar observation was found in ‘Red Delicious’ apples sprayed with ethylene suppressors AVG and 1-MCP resulted in concomitant decreased expression of ethylene biosynthesis genes *MdACS5A* and *MdACO1* in the FAZ and fruit drop [36].

An important aspect of ethylene action in abscission is its perception and tissue sensitivity [8,40]. Ethylene receptors serve as negative regulators to regulated ethylene response, and there is an inverse relationship between receptor levels and ethylene sensitivity of a tissue [41]. The present results show an increase in transcript levels in ethylene receptors *ERS1* and *ETR2-like* in response to hexanal in the FAZ at commercial maturity (Figure 4a; Appendix A). The expression of both receptors could be due to the compensatory mechanism existing within the complex of ethylene receptors. A similar result was observed in AVG-treated nectarine, where *PpETR1* and *PpERS1* transcripts were overexpressed at harvest [42]. Moreover, an increased expression of *Pp-ERS1* was observed in 1-MCP treated peach [43], muskmelon [44], and tomato [45]. Likewise, *LeETR4* was overexpressed in the Never-ripe (*NR*) antisense tomato besides the expected repression of the *NR* transcript [46]. Together, the expression pattern of ethylene biosynthesis and receptor genes in the FAZ prove that improved fruit retention in ‘Honeycrisp’ by hexanal is likely to be ethylene-dependent.

### 3.3. Hexanal Mediates Hormonal Crosstalk in the FAZ

Plant hormones, such as ABA and JA, have a stimulatory effect in abscission [47]. However, when it comes to ABA, whether the stimulatory effect is due to the direct involvement of ABA or mediated by the production of ethylene is still unclear. At commercial maturity, a parallel reduction in the ABA level and expression of ABA biosynthetic related genes *FDPS* and *CLE25* was observed in the hexanal treated FAZ (Figure 4b; Appendix A). Likewise, ABA signalling components *PP2C-34-like*, *PP2C-37*, and ABA response proteins including allergen-like proteins (*Mal d1*, *Mal d1.03G*, *Mal d1-like* and *Mald.06A*) showed decreased expression in the treated FAZ. Earlier studies in melon mature fruit abscission revealed an upregulation of *SnRK2*/PP2Cs that was attributed to early fruit abscission [23], suggesting that the regulation of abscission related signalling compounds trigger the onset of fruit drop and controlling them could lead to preventing such drops.

A key observation by Mou et al. [48] at the onset of tomato fruit ripening revealed a cross-talk between ABA and ethylene, where the authors explained that ethylene might be essential for the induction of ABA biosynthesis and signalling. However, the application of 1-MCP negatively affects this process. The difference of ethylene level in FAZ could not be significantly detected between the control and hexanal treated samples. However, reduced transcripts of key ethylene biosynthesis genes by hexanal in the FAZ may have reduced the ethylene level and thus potentially contributed to a reduction of ABA biosynthesis and signalling in the FAZ, compared to control. In general, ABA and gibberellins (GA) are one pair of classic phytohormones, which antagonistically mediate several plant developmental processes including, fruit-abscission [49,50]. Interestingly, all the DEGs related to GA biosynthesis and signalling were upregulated (Figure 4d; Appendix A), suggesting that hexanal may mediate hormonal cross-talk between ABA and GA. In addition to the above hormones, in plants, melatonin regulates diverse functions including the acceleration of fruit ripening [51]. A proposed role of melatonin in fruit ripening indicated that melatonin acts by upregulating both ethylene and ABA biosynthesis elements, suggesting a cross-talk mechanism between melatonin and other phytohormones [51]. In our study, a significant reduction in melatonin and ABA levels (Table 2), transcripts of ethylene biosynthesis genes by hexanal in the FAZ may have collectively contributed to delay the fruit ripening of treated fruits. However, further research is essential to validate their role in fruit retention.

### 3.4. Hexanal Slows down Cell Wall Degradation and Abscission

The abscission starts with the expression of several wall-loosening enzymes, such as cellulases, polygalacturonases, and expansins. The collective action of all these enzymes accelerates the dissolution of the middle lamella, resulting in organ separation [22,35]. Enlargement of AZ cells involves cell wall loosening, which can be aided by expansins [52]. Several authors have reported that expansins are expressed abundantly in AZ, including tomato flower [53] and apple fruit [54]. The present study has identified seven genes encoding expansins (*EXPs*), which showed decreased expression due to hexanal (Figure 5; Appendix A). Certain AZ cells enlarge in response to ethylene [55]. Microscopic visualization of hexanal treated FAZ cells were smaller, more organized with more defined horizontal layers than control cells (Appendix A). Hexanal presumably reduced ethylene-mediated abscission by the suppression of expansins.

Our results showed that *MdPG1* expression was decreased by hexanal in the FAZ (Figure 5; Appendix A). An increase in polygalacturonase (PG) activity correlates with fruit abscission [54]. *MdPG1* was involved in apple fruit softening, whose expression was reduced by 1-MCP and AVG treatments [36]. An increase in endo-β-1,4-glucanase (EG) activity has been related to fruit abscission in several crops, including apple. Interestingly, all identified EGs were downregulated by hexanal (Figure 5; Appendix A). Moreover, decreased expression of an abscission specific gene, *SAG101-like* (MD17G1039700), encodes an acyl hydrolase involved in senescence/floral organ abscission may assist in retaining the fruits in hexanal treated trees.

In conclusion, this work demonstrates the crucial role of hexanal in improving fruit retention and fruit qualities. The mechanism of improved fruit retention by hexanal in ‘Honeycrisp’ is likely mediated through an ethylene dependent pathway. Hexanal downregulated the ethylene biosynthetic genes in the FAZ and thus may reduce the sensitivity of FAZ cells to ethylene and ABA in the FAZ. Besides, hexanal can maintain the cell wall integrity of abscission zone cells by downregulating cell wall degrading enzymes, such as expansins, EGs, and PG (Figure 7). Thus, hexanal application promises to be a great technology to control fruit drop in ‘Honeycrisp’ apples, given that this cultivar is categorized as more prone to fruit drop. Hexanal is a natural compound produced by all ripening fruits and is generally regarded as safe (GRAS). Moreover, it has been approved by the FDA as a food additive. Further studies could be directed to validate how hexanal slows down the ethylene signal from fruit to the AZ.

## 4. Materials and Methods

### 4.1. Trial Location, Preharvest Treatment and Plant Material Collection

Field trials were conducted at two commercial apple orchards located within the Niagara region of Ontario, Canada (Site A; 43°08′53.7″ N, 79°29′50.2″ W and Site B; 43°11′00.1″ N, 79°34′44.4″ W). Both sites had ‘Honeycrisp’ trees grafted onto M9 rootstocks supported by a trellis system. The trees at Site A were eight years of age, and the trees at Site B were nine years of age. Hexanal formulation (HF) was prepared as described earlier [10,56] containing hexanal at a concentration of 0.02 % (*v*/*v*) in the final spray. ‘Honeycrisp’ trees were subjected to two preharvest sprays of HF approximately four and two weeks before the commercial harvest. Trees sprayed with water served as control. Buffer rows were maintained between the treatments to avoid spray contamination. A total of 48 trees from each site were used for the study.

FAZ samples were collected by cutting about 1 mm at each side of the abscission fracture plane at the base of the pedicel (Figure 8) as described by Zhu et al. [57]. Freshly excised FAZs were flash frozen in liquid nitrogen at the field and stored at −80 °C for hormone analysis and RNA extraction for the RNA-Seq and gene expression studies. FAZ samples were collected only at commercial maturity stage as our intention here is to study the potential effect of hexanal on controlling the genes associated with abscission zone formation at this stage. Fruits that were uniform in both size and development at commercial maturity (starch index 6 and ground color change from green to yellow) and 42 days after commercial maturity were also harvested and immediately brought to the laboratory for the quality traits measurements.

### 4.2. Fruit Retention and Fruit Quality Measurements

Four trees with uniform growth, similar maturity, comparable fruit count, and similar location for wind direction were marked for fruit retention study. Fruit retention (FR) was monitored on a biweekly basis from the commercial harvesting in September to before the first frost in November and expressed as a percentage using the following formula:Fruit Retention % = 100 − [(initial fruit count − final fruit count)/initial fruit count) × 100]

Ten randomly selected, similar-sized fruits at commercial maturity (0 days) and end of the fruit retention study (49th day) were used for the quality measurements. Fresh weight (g) was measured using an electronic balance. Two firmness readings (N) were taken using a handheld penetrometer with an 11-mm diameter tip (Effegi pressure tester, Via Reale, 63, Facchini 48011, Alfonsine, Italy) on the opposite sides of each fruit. Two vertical slices from each side of the apples were freshly juiced, and TSS (°Brix) readings were measured using a prism refractometer (Fisher Scientific, Mississauga, ON, Canada).

### 4.3. Plant Hormone Measurement

Eight randomly selected apples were used for the ethylene measurement. Fruits were weighed and placed in 2 L glass bottles. Bottles were sealed for an hour with a lid containing a rubber port where a syringe was used to collect 1 mL of headspace gas after gently shaking the bottles to mix up the air inside. The gas sample was immediately injected into an SRI-8610c gas chromatograph equipped with a 0.5 mL sample loop, and ethylene was detected using a flame ionization detector (Varian Inc., Mississauga, ON, Canada).

Plant hormones present in the FAZ (Figure 8) were extracted using the methanol double extraction method. Briefly, 25 mg of freeze-dried, powdered FAZ samples were extracted with a solvent (methanol:formic acid:milli-Q H_2_O = 15:1:4), and the homogenate was kept at −20 °C for an hour. The supernatant was then collected through centrifugation (15 min, 14,000 rpm). The pellet was re-extracted using the same protocol, and the supernatants were pooled. The pooled supernatant was then evaporated to dryness using nitrogen gas in a fume hood. The dried samples were reconstituted using a buffer solution (0.1% formic acid:acetonitrile = 97:3), then filtered through a 0.45-µm centrifuge filter (Millipore; 1 min, 13,000 rpm). The supernatant was then transferred to a 96-well collection plate. Metabolites were separated by reverse-phase ultra-performance liquid chromatography (UPLC) system with detection using an Aquity QDa single quadruple mass spectrometer (MS) controlled by Empower 3 (Waters limited, Mississauga, ON, Canada) by injecting a 5 μL aliquot of sample onto an Acquity BEH Column (2.1 × 50 mm, i.d. 2.1 mm, 1.7 μm). Metabolite peaks were monitored in single ion recording mode and quantified using a standard curve [58].

### 4.4. RNA-Isolation, Library Preparation and Sequencing

Total RNA was extracted from FAZ tissues using an RNA isolation kit (Norgen Biotek, Thorold, ON, Canada). RNA quality was verified and quantified using Nanodrop^TM^ (2000/2000c Spectrophotometers, Thermo Fisher Scientific, Wilmington, DE, USA). One microgram of mRNA was used as a template for first-strand cDNA synthesis using NEBNext^®^ Poly(A) kit (NEB #E7490, New England Biolabs, Inc., Ipswich, MA, USA) and NEBNext^®^ ultra™ II directional RNA library prep kit for Illumina (NEB #E7760, New England Biolabs, Inc., Ipswich, MA, USA). Paired-end sequencing (75 bp) was performed for four samples using NextSeq 500/550-mid output kit v2.5 (2 × 75 cycles) on an Illumina NextSeq500 sequencer (Norgen Biotek, Thorold, ON, Canada).

### 4.5. Trimming, Assembly, and Annotation of Paired-End Sequenced Reads

The quality of raw sequences was measured with FastQC (v 0.11.9) using per base and sequence GC content [59] and were trimmed by Trimmomatic (v.0.36) using default parameters [60]. Then adapter sequences used for library preparation were removed with Cutadapt (v 2.8) using sequencing by oligonucleotide ligation and detection colour space algorithm (SOLiD) [61]. Trimmed reads were assembled with STAR aligner (v2.1.3) with default parameters. Apple genome project GDDH13, version1.1 was used as a reference genome [62]. FeatureCounts program (v 1.22.2) was used to assign sequence reads to each gene in the samples corresponding to the apple reference genome, GDDH13, version 1.1. A table of count reads was created with rows corresponding to genes and columns to samples [63].

### 4.6. Differentially Expressed Gene Analysis

Empirical analysis of digital gene expression data—EdgeR [64], an R Bioconductor package deposited in the DEBrowser (v1.16.1) [65], was used to analyze the differentially expressed genes (DEGs) between hexanal treated and control samples at *p* ≤ 0.05 and gene expression fold change ≥ 2. The EdgeR models count data using an over-dispersed Poisson model and an empirical Bayes procedure to moderate the degree of overdispersion across genes. The table of count reads of the samples was fed to the EdgeR program (v 3.14.0). DEGs were analyzed through data assessment, normalization, and DEG detection using EdgeR models. The EgdeR modelled data to negative binomial (NB) distribution, Y_gs_~NB(X_s_Z_gs_, ɸ_g_) for gene g and sample s. Here X_s_ is the library size, ɸ_g_ is the dispersion, and Z_gs_ is the relative abundance of gene g into which sample s belongs. The NB distribution reduces to Poisson when ɸ_g_ = 0 [64]. M values Trimmed mean of M-values (TMM) normalization method was used to normalize the reads counts during internal modelling of the samples using EdgeR package [64].

### 4.7. Enrichment Analyses

Gene ontology (GO) and functional pathway enrichment analyses were performed in ShinyGO (v0.61), based on hypergeometric distribution followed by Benjamini–Hochberg correction with a false discovery rate (FDR) at *p* ≤ 0.05 [66]. Three different gene ontologies, i.e., biological process (BP), molecular function (MF), and cellular components (CC), were analyzed separately. The relationship between enriched functional pathways was visualized using an interactive plot. A hierarchical clustering tree was also used to summarize the correlation among the enriched pathways. *Arabidopsis thaliana* (TAIR10) ortholog genes for the identified *Malus domestica* DEGs (726) were retrieved from the phytozome database (v13.1.6) and were used for the enrichment analysis [67]. Query-based gene information of all detected DEGs was obtained using the phytozome database for the *Malus domestica* (GDDH13, v1.1).

### 4.8. Quantitative RT-PCR

Quantitative reverse transcription PCR was conducted for eight genes chosen to represent ethylene biosynthesis and signalling pathway and cell wall modification. Gene-specific primers were designed using Primer3Plus software. Two micrograms of total RNA extracted from FAZ were reverse transcribed with Superscript II reverse transcriptase (Invitrogen, Burlington, ON, Canada). qPCR reactions were performed in 10 μL, containing 5 μL SYBR^®^ Green Supermixes (Invitrogen, Burlington, ON, Canada), 50 ng of cDNA and 2.5 μL of 400 nM of each primer (Appendix A). Four biological and three technical replicates for each gene were analyzed using a CFX96 Real-Time PCR detection system (BioRad, Mississauga, ON, Canada). *Malus domestica Actin* (*MdACT*) and *Histone-3* (*MdHIS-3*) genes were used as reference genes to normalize the gene expression of a target gene. The gene expression was quantified using the 2^−ΔΔCt^ method [37].

### 4.9. Statistical Analysis

Fruit retention, and hormone data were analyzed using general linear mixed models (proc GLIMMIX) in SAS v9.4 (SAS Institute, Raleigh, NC, USA). Variances of fixed effects, such as location and treatment, were partitioned from random effects, which include replication. Shapiro–Wilk normality tests and studentized residual plots were used to test error assumptions of variance analysis, including random, homogenous, and normal distributions of error. Means were calculated using the LSMEANS statement, and significant differences between the treatments were determined using a post-hoc Tukey–Kramer HSD test with α = 0.05 and are mentioned in each figure or table.

## Figures and Tables

**Figure 1 ijms-22-08830-f001:**
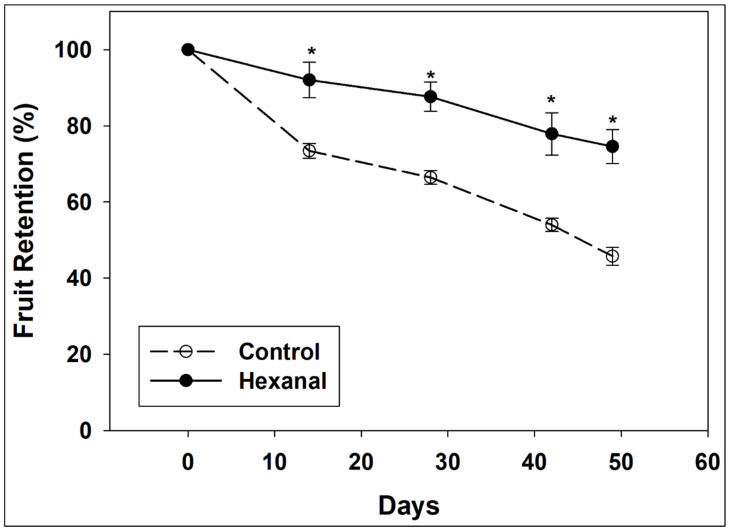
Percentage of fruit retention in control and hexanal treated ‘Honeycrisp’ trees throughout the 49 days of study period. The fruits are at commercial maturity at 0th day and on the same day, grower started harvesting apples from other trees in the same orchard. Each value represents the mean ± SE of 4 trees. Asterisks indicate significant differences between control and hexanal treatment at the same sampling time based on Tukey’s HSD test at α = 0.05.

**Figure 2 ijms-22-08830-f002:**
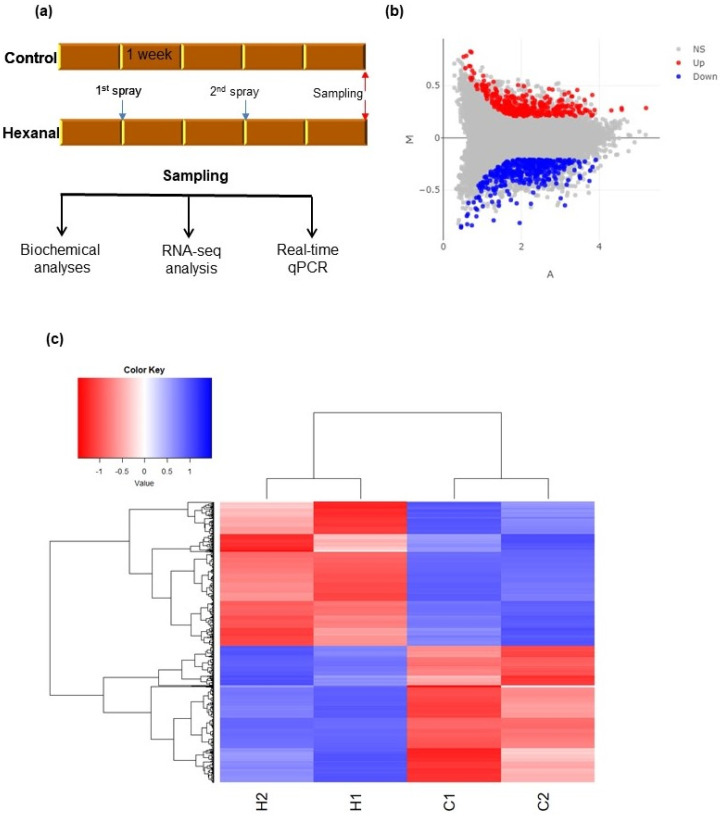
Identification of the differentially expressed genes (DEGs) between control and hexanal treated FAZ samples. (**a**) layout of the overall experimental procedure where samples for the RNA-seq analysis were collected at commercial maturity; (**b**) represents the MA plot shows the relationship between the expression change (M) and average expression strength (A) of the 353 up (red) and 373 down regulated genes (blue). Genes that pass a threshold of *p* ≤ 0.05 and |log_2_foldchange|> 1 in differential expression analysis are considered as upregulated. Whereas genes pass a threshold of *p* ≤ 0.05 and|log_2_foldchange|> −1 are considered as downregulated. If any gene did not meet the above requirements are considered as non-significant (NS); (**c**) heat map of the 726 DEGs shows the variation across the four samples harvested from two sites.H1 and H2 represent the hexanal treated samples harvested from site A and B, respectively. Similarly, C1 and C2 represent the control samples harvested from site A and B, respectively. Additional information about the DEGs is presented in Appendix A.

**Figure 3 ijms-22-08830-f003:**
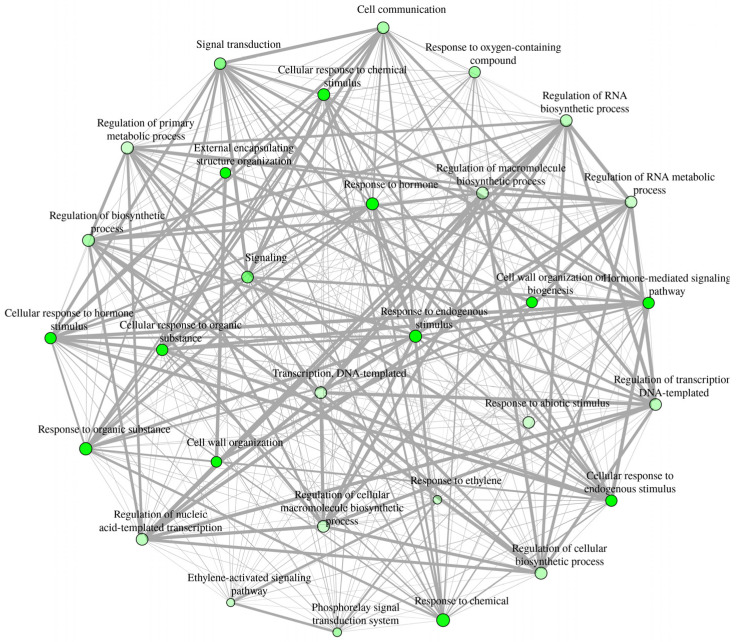
Interactive enrichment networks plot of first 30 enriched functional categories belonged to BP (FDR, *p* < 0.001, at edge cutoff 0). In the network analysis, two pathways (nodes) are connected if they share 20% (default) or more genes. Darker nodes are more significantly enriched gene sets. Bigger nodes represent larger gene sets. Thicker edges represent more overlapped genes. A hierarchical clustering tree summarizing the correlation among these top 30 significant pathways was included in Appendix A. Additional information about the functional pathways is presented in Appendix A.

**Figure 4 ijms-22-08830-f004:**
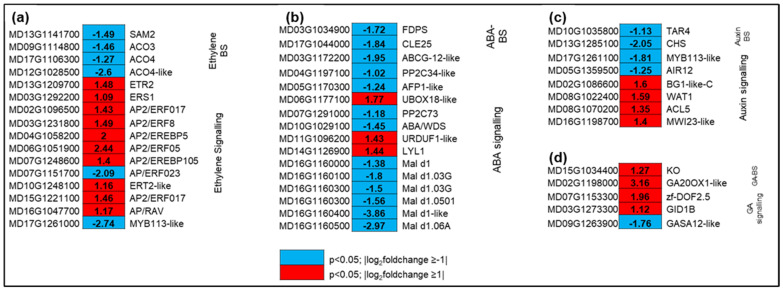
Expression profiling of genes related to biosynthesis and signaling of plant hormones (**a**) ethylene, (**b**) ABA, (**c**) auxin, and (**d**) GA. Blue and red represent downand upregulated gene expression due to hexanal application in the FAZ at harvest. The left column shows the *Malus domestica* gene id, the middle column shows the gene expression with |log_2_foldchange| values, and the right column shows the corresponding gene id. Additional information on the hormone-related genes is presented in Appendix A.

**Figure 5 ijms-22-08830-f005:**
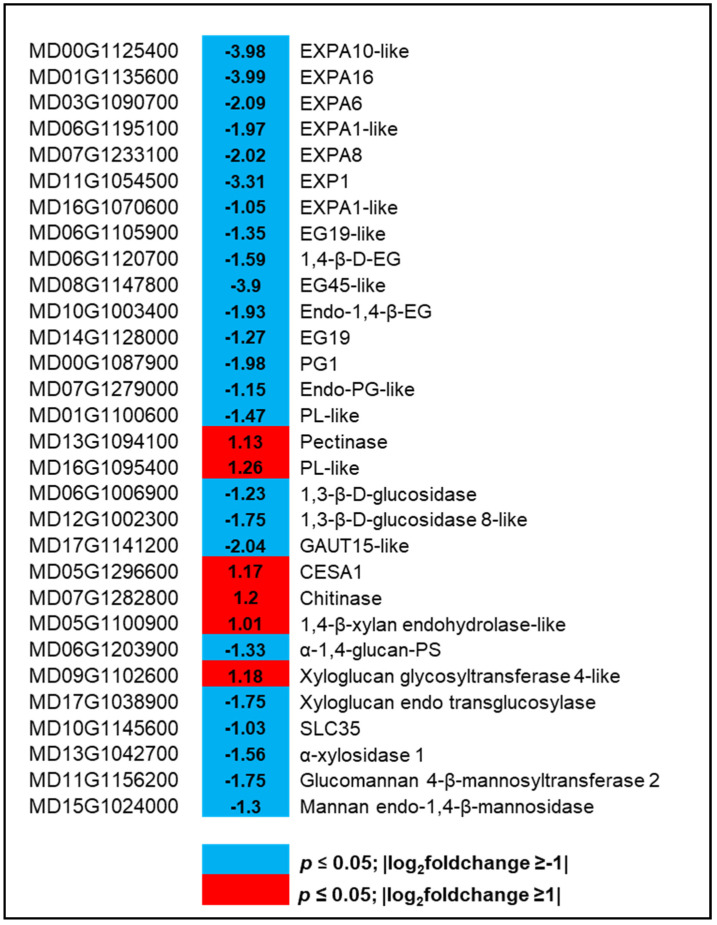
Expression profiling of genes related to cell wall modification. Blue and red represent down-and upregulated gene expression due to hexanal application in the FAZ at harvest. The left column shows the *Malus domestica* gene id, the middle column shows the gene expression with |log_2_foldchange| values, and the right column shows the corresponding gene id. Additional information on the cell wall modification genes is presented in Appendix A.

**Figure 6 ijms-22-08830-f006:**
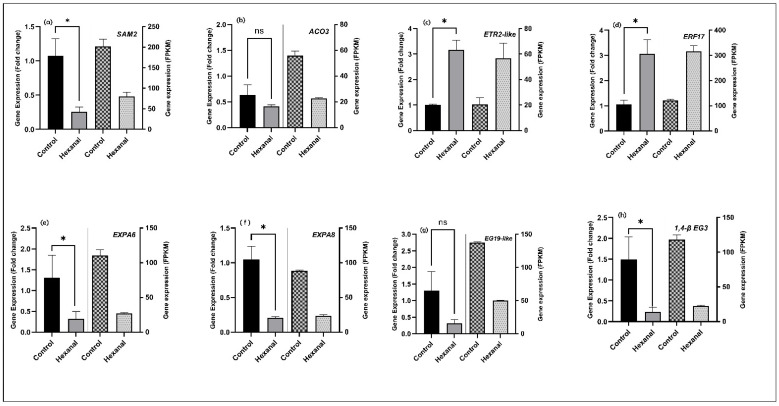
qRT-PCR confirmation of gene expression pattern of selected eight genes representing (**a**–**d**) ethylene biosynthesis and signalling and (**e**–**h**) cell wall modification. The data represents the mean ± SE of four biological replicates and three technical replicates representing the samples harvested at commercial maturity from both commercial orchards. Fold change values were calculated based on 2^−ΔΔCt^ method by Livak, and Schmittgen [37]. Means followed by asterisks indicate significant differences between control and hexanal formulation treatment based on unpaired *t*-test with Welch’s correction at α = 0.05. FPKM values of each gene were calculated from RNA-Seq reads counts normalized to a per million total reads counts. Genes and the primers are shown in the Appendix A.

**Figure 7 ijms-22-08830-f007:**
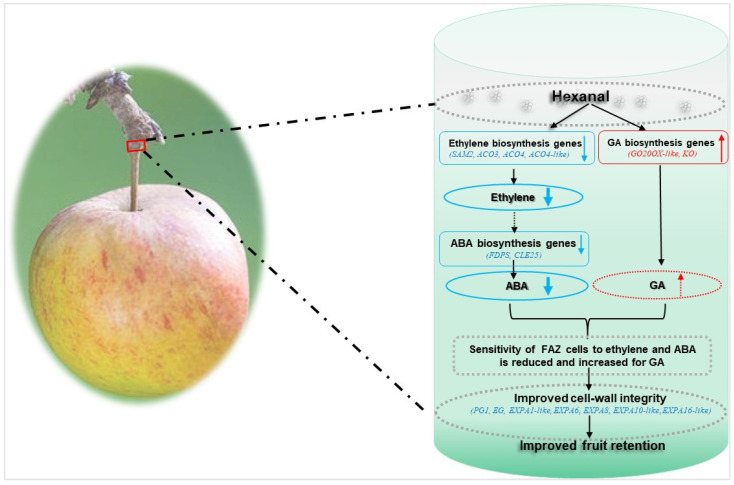
Proposed model of hexanal improved fruit retention in ‘Honeycrisp’ apples. Preharvest hexanal spray downregulated the expression of genes involved in ethylene biosynthesis in the FAZ and thus decreased the ethylene. Lower ethylene, in turn slows down the expression of the ABA biosynthesis genes and substantially minimize the ABA level in the FAZ. At the same time, GA biosynthesis genes were upregulated by hexanal and may enhance the GA concentration. Hence, the sensitivity of FAZ cells to ABA decreased. Parallelly, hexanal also downregulated genes related to cell wall degrading enzymes such as EG, PG, and expansins. Consequently, cell wall integrity of the FAZ cells improved in the treated fruits. Collectively, these events improved the fruit retention of the hexanal treated fruits. “Solid arrows represent known mechanism; broken arrows represent unknown mechanism; blue represent downregulation/decrease events, red represent upregulation/increase events”.

**Figure 8 ijms-22-08830-f008:**
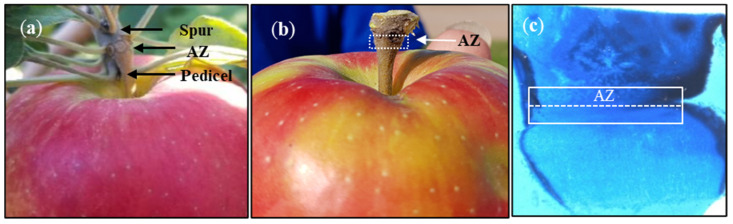
Anatomical observation of the fruit abscission zone (FAZ) of ‘Honeycrisp’ apple. (**a**) photograph shows the abscission zone (AZ) of the fruit located between the spur and pedicel of the fruit stalk; (**b**) photograph shows about 1 mm size of the AZ was manually dissected with a razer blade at each side of the abscission fracture plane; (**c**) microscopic view of the AZ region. The AZ looks like a funnel shape with constrictions in both sites. The broken line indicates the position of the abscission fracture plane. The AZ was stained using lactophenol cotton blue.

**Table 1 ijms-22-08830-t001:** Fruit quality parameters of control and hexanal treated ‘Honeycrisp’ apples at commercial maturity (0th day) and 49 days after commercial maturity.

Parameter	Treatment	Days (d)	
		**0**	**49**
Firmness (N)	Control	76.00 ± 1.13	65.35 ± 1.22
	HF	78.63 ± 0.84	71.46 ± 1.20 *
TSS (°Brix)	Control	13.47 ± 0.17	14.92 ± 0.11
	HF	13.37 ± 0.16	15.07 ± 0.16
Fresh weight (g)	Control	209.30 ± 8.98	237.8 ± 15.5
	HF	239.10 ± 15.37	250.20 ± 9.21

Values represent the mean ± SE of 10 randomly selected fruits. Means followed by asterisks indicate significant differences between control and hexanal treatment at the same sampling time based on student’s *t* test at *p* < 0.05.

**Table 2 ijms-22-08830-t002:** Variation of plant hormones in control and hexanal treated ‘Honeycrisp’ fruit and in fruit abscission zone (FAZ).

Treatment	Ethylene	Abscisic Acid	Zeatin	Melatonin
	(μL·L^−1^·kg^−1^·h^−1^)	(ng. g^−1^, DW)	(ng. g^−1^, DW)	(ng. g^−1^, DW)
Control	5.38 ± 0.88	320.17 ± 33.25	465.23 ± 45.65	56.58 ± 6.37
Hexanal	4.24 ± 0.53	192.99 ± 11.83 *	441.03 ± 16.53	39.81 ± 2.47 *

Ethylene was measured in randomly selected eight fruits per treatment, and other plant hormones represent the mean ± SE of 18 replications of FAZ tissues harvested from two commercial orchards. Means followed by asterisks indicate significant differences between control and hexanal treatment based on Tukey’s HSD test at α = 0.05.

## Data Availability

All data supporting the findings of this study are available within the paper and within its Appendix A published online.

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
