# Peer review of "Transcriptomics of Improved Fruit Retention by Hexanal in ‘Honeycrisp’ Reveals Hormonal Crosstalk and Reduced Cell Wall Degradation in the Fruit Abscission Zone"

_ijms, 2021, doi:10.3390/ijms22168830_

Round 1
Reviewer 1 Report
This manuscript presents the effects of hexanal on fruit retention, fruit firmness, hormone levels in FAZ, and the expression of hormone-related genes in FAZ. Authors try to make a hormonal crosstalk. However, in several parts of the manuscript, the description of the ethylene data is not accurate. The title is not clearly written. Besides “fruit retention”, “improved fruit firmness” is also an important element in the manuscript. This element should be mentioned in the title. The saying of “reduced cell-wall remodeling” cannot bring forth the idea of “maintained cell-wall integrity” (line 27). The “fruit retention” data is important for the manuscript. However, the meaning of “postharvest fruit retention” is confusing. Authors should improve the accuracy of data description and adjust the discussion on ethylene. The gene expression level was studied in the FAZ. The discussion on fruit firmness is only supported by the fruit ethylene level. However, in Table 2, it was shown that hexanal did not significantly altered the fruit ethylene level. Authors should consider to tone-down the manuscript to only discuss the effects of hexanal on fruit retention.
Detailed comments are listed below:
Line 22: “involve”->”involved”
Line 38: To enrich the background of the research, authors should give more examples of the factors other than climate.
Line 42: “Pre-harvest fruit drop” -> “PFD”.
Lines 33-34: cite a reference for “ranks third in global fruit production”
Line 55: cite references to support the role of ethylene on abscission
Lines 57-59: need to be supported by references
There is not a “Figure 1” before “Figure 2”
Figure 2: the statistical test used should be mentioned in the Figure legend.
Lines 107-110: the meaning of “postharvest” is unclear. The previous paragraphs talk about “Pre-harvest” but suddenly, in line 107, the description is changed to “postharvest”.
Figure 2: what is the meaning of “postharvest fruit retention”? After harvest, why are the fruits retained?
Line 110: the meaning of “days after postharvest” is unclear.
Line 147: “in” fruit abscission zone?
Table 2: Hexanal treatment led to the change of melatonin level. However, in the whole manuscript, melatonin is not discussed. If authors want to keep this data, authors should address how the melatonin level relates to hexanal and the fruit property.
Line 146: the model of the HPLC-MS system should also be mentioned.
Lines 151-152: the statistical test used should be mentioned in the figure legend.
Figure 3a: in the figure legend, the meanings of the blocks and the different colors should be explained. According to the position of the arrows, the samplings of the control group and the hexanal group were conducted at different time points. The comparison between the two groups would be unfair.
Figure 3: remove meaningless symbols such as the symbol next to “353” in Figure 3b.
Figure 3b is unnecessary and misleading.
Figure 3d: the resolution is too low. The labels cannot be seen clearly.
Figure 3c: the meanings of “NS”, “Up” and “Down” should be explained in the figure legend.
Line 203: sixty-one -> 61
Line 215: sixteen -> 16
Line 310: what do the error bars represent?
Figure 7: in the figure legend, the calculation method for fold change determination of the qPCR results should be explained. Are the two groups statistically different?
Line 317: the claim “hexanal treated fruits produced less ethylene” is not supported by the data. In Table 2, it is shown that the ethylene level was not statistically different between the hexanal treated group and the control group.
Line 366: the hexanal-mediated change of ethylene level is not supported by the data.
Lines 395-396: the hexanal-mediated decrease of ethylene production is not supported by the data. Fruit retention is more likely related to the abscission of FAZ. In FAZ, genes related to hormones other than ethylene are regulated by hexanal treatment. Why do authors only refer the mechanism to ethylene?
Author Response
Response to reviewer’s comments for ijms-1295635, “Transcriptomics of improved fruit retention by hexanal in ‘Honeycrisp’ reveals hormonal crosstalk and reduced cell-wall remodelling in the fruit-abscission zone.”
The reviewer’s questions have been repeated (underlined), and our response follows. We sincerely thank both the reviewers and editor for their assistance with this manuscript.
SUMMARY
We have made corrections following the two reviewers’ suggestions. In summary, we edited the manuscript in the suggested places, from the introduction to the conclusion sections of the manuscript.
REVIEWER 1
- Line 22: “involve”-> ”involved”
Thank you for notifying the grammatical error. We have changed the word “involve” to “involved” in the text (Line 22)
- Line 38: To enrich the background of the research, authors should give more examples of factors other than climate.
We have included the additional factors that influence preharvest fruit drop in apples. These include orchard, climate, and fruit factors such as mineral nutrients, summer pruning, water availability, growing season temperatures [Robinson, 2011], lower starch content and higher internal ethylene concentration [Greene, 2013]. (Line 38-40) The details of the references are below.
[2] Robinson, T. The physiology of apple pre-harvest fruit drop. URL: http://www.hort.cornell.edu/expo/proceedings/2011/Management%20of%20Pre-Harvest%20Fruit%20Drop/Physiology%20of%20fruit%20drop.pdf. 2011. (accessed on 20 July 2021).
[4] Greene, D. W., Krupa, J., & Autio, W. Factors influencing preharvest drop of apples. In XII International Symposium on Plant Bioregulators in Fruit Production. 2013, 1042 (pp. 231-235).
- Line 42: “Pre-harvest fruit drop” -> “PFD”.
“Pre-harvest fruit drop” has been changed to “PFD” (Line 45)
- Lines 33-34: cite a reference for “ranks third in global fruit production”
We have included below reference to support that apple ranks third in global fruit production.
[1] Statista, Global fruit production in 2019, by selected variety (in million metric tons)* https://www.statista.com/statistics/264001/worldwide-production-of-fruit-by-variety/. 2021, (Line 34)
- Line 55: cite references to support the role of ethylene on abscission
We have included two key references such Osborne, 1989 and Meir et al., 2006 to support the role of ethylene on abscission. The details of the references are below.
[17] Osborne, D. J. Abscission. Crit Rev Plant Sci. 1989, 8:103–129
[18] Meir, S.; Hunter, D. A.; Chen, J. C.; Halaly, V.; Reid, M. S. Molecular changes occurring during the acquisition of abscission competence following lowering auxin depletion in Mirabilis jalapa. Plant Physiol. 2006, 141:1604–1616
- Lines 57-59: need to be supported by references
We have included two key references, such as Yuan, 2007 and Cline, 2019 to support that ethylene releasing compound, ethephon, promoted fruit abscission in ‘Golden Delicious’ and ‘Honeycrisp’ apples. The details of the references are below.
[20] Yuan, R. Effects of temperature on fruit thinning with ethephon in ‘Golden Delicious’ apples. Sci. Hortic. 2007, 113(1), 8-12.
[21] Cline, J. A. Multiple season-long sprays of ethephon or NAA combined with calcium chloride on ‘Honeycrisp’apples: I. Effect on bloom and fruit quality attributes. Can. J. Plant Sci. 2019, 99(4), 444-459.
- There is not a “Figure 1” before “Figure 2”
Figure 1 was included in the materials and method section under the “Trial location, preharvest treatment and plant material collection” sub-section. (Line 439)
- Figure 2: the statistical test used should be mentioned in the Figure legend
The significant differences between control and hexanal treatment at the same sampling time were analyzed based on a post-hoc Tukey’s HSD test at α = 0.05. The statistical test has been mentioned in the figure legend. (Line 123)
We have included the statistical test in all other figures and tables. (Line 135, 157, 305-306)
- Lines 107-110: the meaning of “postharvest” is unclear. The previous paragraphs talk about “Pre-harvest,” but suddenly, in line107, the description is changed to “postharvest”.
We are sorry for the confusion of the “days postharvest.” Briefly, we selected trees that are similar in size, have comparable fruit count and maturity. The fruits of the selected trees were kept unpicked throughout the fruit retention study. We counted the fruits retained on the trees from the stage of commercial maturity, which is mentioned as the 0th day in figure 2. On the day it happened, the grower started harvesting apples from other trees in the same orchard. Therefore, fruit counting days beyond this point were considered as “days-postharvest.” However, to avoid confusion, we have deleted the word “postharvest” from the text, figure 2 and table 1. Instead, we mentioned it as “days” (Line 111, 114, 120, 121, 133). A new figure was replaced for figure 2.
- Figure 2: what is the meaning of “postharvest fruit retention”? After harvest, why are the fruits retained?
We are sorry again for the confusion. It is fruit retention. To avoid confusion, we have deleted this phrase from the main text, figure 2 and table 1. (Line 111, 114, 120, 121, 133)
- Line 110: the meaning of “days after postharvest” is unclear.
We are sorry for the confusion. We have deleted this phrase from the main text, figure 2 and table 1. (Line 111, 114, 120, 121, 133)
- Line 147: “in” fruit abscission zone?
Thank you for notifying this. We have included the preposition “in” in the title of table 2. (Line 154)
- Table 2: Hexanal treatment led to the change of melatonin level. However, in the whole manuscript, melatonin is not discussed. If authors want to keep this data, authors should address how the melatonin level relates to hexanal and the fruit property.
Yes, we agree that the melatonin level was significantly decreased by hexanal treatment. Several reports suggested that melatonin enhances the senescence process by regulating ethylene and ABA in fruit crops (Arnao and Ruiz, 2020). However, we could not detect any genes/enriched functional GO terms and pathways associated with melatonin. Therefore, we were reluctant to speculate how hexanal modulated melatonin levels using only the hormonal data. This could be an interesting direction in the future for us to specifically focus on how hexanal modulates melatonin in the fruit retention aspect.
- Line 146: the model of the HPLC-MS system should also be mentioned.
The reverse-phase ultra-performance liquid chromatography (UPLC) with Aquity QDa single quadruple mass spectrometer (MS) controlled by Empower 3 (AQUITY, Waters, Canada) was used for the hormone analysis. The model of the UPLC-MS has been included in Line 151-153.
- Lines 151-152: the statistical test used should be mentioned in the figure legend.
As previously mentioned, the statistical test (Tukey’s HSD test at α = 0.05) was included in the table footnote (Line 159)
- Figure 3a: in the figure legend, the meanings of the blocks and the different colors should be explained. According to the position of the arrows, the samplings of the control group and the hexanal group were conducted at different time points. The comparison between the two groups would be unfair.
The blocks are showed in two different colours only to indicate the progression of fruit colour towards maturity. However, to avoid confusion, we have removed the colours. We are sorry that the arrows used to indicate the sampling time were slightly moved in two different positions in figure 3. A sampling of both control and hexanal-treated fruits was done on the same day, during morning hours, before temperature rise. All samples were promptly frozen in Liq. N2 at the field and stored at -80 oC until further analysis. We have mentioned it in the materials and method section (Line 445-459). The previous figure 3 was replaced by new figure 3. (Line 172)
- Figure 3: remove meaningless symbols such as the symbol next to “353” in Figure 3b.
The symbol was removed from figure 3b. (Line 172)
- Figure 3b is unnecessary and misleading.
Figure 3b was removed from the main figure 3. However, we have indicated that 353 differentially expressed genes were upregulated while 373 genes were down-regulated between control and hexanal treatment at commercial maturity. (Line 174)
- Figure 3d: the resolution is too low. The labels cannot be seen clearly.
The figure 3d, the heat map was adjusted to meet the image requirement. (Line 174)
- Figure 3c: the meanings of “NS”, “Up” and “Down” should be explained in the figure legend.
Genes that pass a threshold of padj < 0.05 and |log2foldchange|> 1 in differential expression analysis are considered as up regulated (red). Whereas genes pass a threshold of padj < 0.05 and |log2foldchange|> -1 are considered as downregulated (blue). If any gene did not meet the above requirements are considered as non-significant (NS). The explanation has been added in the figure 3 legend. (Line 175-188)
- Line 203: sixty-one -> 61
sixty-one is replaced by 61
- Line 215: sixteen -> 16
sixteen is replaced by 16
- Line 310: what do the error bars represent?
In figure 7, the error bar represents the standard error values of four biological replicates used for the gene expression studies. In FPKM values, the error bar represents the two replications of the RNA-Seq read counts normalized to a per million total reads counts. We have included the description in figure 7. (Line 312-318)
- Figure 7: in the figure legend, the calculation method for fold change determination of the qPCR results should be explained. Are the two groups statistically different?
The gene expression fold change of the qPCR results was calculated using the 2-ΔΔCt method by Livak and Schmittgen, 2001. The expression fold change of control and hexanal treatment are statistically different (P < 0.05) in all genes except ACO3 and EG19-like. We have included a new graph that shows the statistical significance between the control and treated groups (figure 7). The calculation and significance were included in the legend of figure 7. (Line 312-318)
- Line 317: the claim “hexanal treated fruits produced less ethylene” is not supported by the data. In Table 2, it is shown that the ethylene level was not statistically different between the hexanal treated group and the control group.
We agree with your statement that the ethylene production value was not statistically differed (P < 0.05) between control and treated fruits at commercial maturity. However, it was significant at p <0.1. Although the preharvest fruit drop in ‘Honeycrisp,’ usually begins 3-4 weeks before harvest, a substantial portion of fruits started to drop during the commercial harvesting period, according to our growers and Arseneault and Cline (2017). By this time, it has been observed that ethylene production also rises substantially (Robinsion, 2011; Li et al., 2010). Our fruit retention results also showed that a substantial portion of control fruits have dropped during the commercial harvesting period. In our study, the nature of fruit drop and ethylene production data in the control fruits are in agreement with previous observations. On the contrary, treated trees that retained more fruits during this period may have been associated less ethylene production in the fruits than that of control fruits. (Line 334)
- Line 366: the hexanal-mediated change of ethylene level is not supported by the data.
Despite the fact that the ethylene production data in the fruits did not show a statistical difference between control and treated fruits at P 0.05, possibly in part due to the methodology used, which was unable to assess ethylene production in the fruit abscission zone (FAZ), as well as the difficulty in determining whether the measured ethylene production was due to ripening or injury during FAZ tissue dissection. So to address it, gene expression of the etyhlene biosynthesis genes was quantiified. For instance, key genes involved in the ethylene biosynthesis pathway (SAM2, ACO3, ACO4 and ACO4-like) were identified, and all were significantly down-regulated by hexanal. In addition, two receptor genes that are acting as a negative regulator in the ethylene signalling pathway were identified, and both were significantly upregulated by hexanal (ETR2 and ERS1). We claimed that hexanal mediates changes in ethylene level based on the gene expression variation information in the FAZ and ethylene production by fruits. (Line 381).
- Lines 395-396: the hexanal-mediated decrease of ethylene production is not supported by the data. Fruit retention is more likely related to the abscission of FAZ. In FAZ, genes related to hormones other than ethylene are regulated by hexanal treatment. Why do authors only refer to the mechanism to ethylene?
We greatly appreciate the question. We do agree that other hormones and their respective genes were also identified in the FAZ, and we mentioned that hormonal variation in the discussion (Line 370-387). However, among the hormonal gene expression variation identified by the RNA-Seq data, we have identified many enriched GO terms functional classes (supplementary data S3) and genes (16 out of 61) associated with ethylene were altered by hexanal. In addition, we have identified 31 genes related to cell-wall modification, which are generally altered by ethylene in climacteric fruits. Ethylene production rises during commercial maturity in varieties like ‘Honeycrisp’, which is identified as a “medium” level ethylene production variety among the apples. Therefore, our discussion focused more on the ethylene mechanism by having the DEGs and enriched functional classes and pathways.
Reviewer 2 Report
Sriskantharajah et al. describe a study to examine the effects of hexanal spray application on physiological characteristics believed contributing to pre-harvest fruit drop in apples (particularly, the variety Honeycrisp). This manuscript is well organized and well written and was an interesting read. In light of this, I have only comments and suggestions for improvement of the text (e.g. Figure 4), and a few comments about the nature of the specificity of the inferences being made as far as gene expression in the fruit abscission zone (see below).
Main points:
1. Inferences about gene expression in the fruit abscission zone: My main point of concern about the manuscript is the inferences regarding the FAZ. From the Methods, it is clear that during tissue collection from the region around the true FAZ a portion of the spur and fruit pedicel are also collected. Given this was done by hand, in the field, there was likely variance in how much of each tissue was collected from each sample. What I am getting at of course is there would be a variable signal of differential gene expression (and hormone production) unique to the spur tissue and pedicel tissue that would be included in your analysis. In order for a proper control you would need to have a separate DE analysis for the spur and pedicel and then try and find genes regulated differently in the sample containing the FAZ.
My question then is : should you be making the specific inference that the FAZ alone has these DE genes? In my opinion you likely will need to scale back your conclusions about the FAZ and be more general about the effect of hexanal on gene expression on the entire region of the spur-pedicel interface.
2. Timing of sampling relative to hexanal application: Similarly to point one above, I think it is worth putting into context in the discussion that the DE analysis described here is a single point in time relative to the hexanal application. This snapshot in time does not really say much about the rate of change of expression from the time of hexanal application (i.e. was the change in expression immediate? Or did it ramp up slowly? Or is what we observed greatly diminished from some maximum). Clearly having more time points in the analysis would clear up this uncertainty.
3. Figure 4: In my opinion Figure 4 “as is” should not be included in the main text of the manuscript. I feel that parts “a” and “c” can easily be explained as text, and part “b” is much too cluttered. The text is cut off in some of the GO names, for example. I think this network “hairball” could be an interesting figure if it can be simplified to perhaps only including the top 20(?) GO categories.
4. Figure 8: Figure 8 is quite well done, and perhaps would be the jumping off point to say that this really is the proposed model if the FAZ really does have the observed gene DE. With the caveats raised in point 1 above.
5. Consistency with software names/versions: There are a few instances of software being mentioned without a reference or version number. In addition, I suggest you mention whether default software parameters were used during analysis.
6. DE normalization and EdgeR: One thing that jumped out at me, having not used EdgeR before, was the lack of details about the read normalization for the DE analysis. I gather this normalization is carried out during its internal modelling, but it is not clear form the text this is the case. A quick mention of normalization would help the reader here, since anyone mildly familiar with DE analysis will be looking for something of this nature in the text.
7. Maturity of the sampled fruit: There are several mentions about the sampling of the fruit and fruit maturity in the Methods. It is not clear however what criteria was used to assume fruit maturity. Starch/Sugar test?, Fruit colour?
8. Fruit retention and fruit weight: It is not clear from the text whether any measurements were made to see if avg. fruit weight (of dropped fruit) of the test trees was different, leading to confounding of results of the fruit retention analysis.
Author Response
REVIEWER 2
- Inferences about gene expression in the fruit abscission zone: My main point of concern about the manuscript is the inferences regarding the FAZ. From the Methods, it is clear that during tissue collection from the region around the true FAZ a portion of the spur and fruit pedicel are also collected. Given this was done by hand, in the field, there was likely variance in how much of each tissue was collected from each sample. What I am getting at of course is there would be a variable signal of differential gene expression (and hormone production) unique to the spur tissue and pedicel tissue that would be included in your analysis. In order for a proper control, you would need to have a separate DE analysis for the spur and pedicel and then try and find genes regulated differently in the sample containing the FAZ.
My question then is: should you be making the specific inference that the FAZ alone has these DE genes? In my opinion you likely will need to scale back your conclusions about the FAZ and be more general about the effect of hexanal on gene expression on the entire region of the spur-pedicel interface.
Thank you for the question. We agree with your concern, and we believe that figure 1 (a, b) might have caused this confusion. We are sorry that the fruit abscission zone (FAZ) area was not marked in picture 1b. We harvested the tissue in the constriction region by cutting about 1-1.5 mm at each side of the abscission fracture plane at the base of the pedicel. We were very careful in not disturbing the spur and avoided collecting the FAZ from fruits that had a short pedicel that smudges the demarcation between spur and pedicel. We followed a similar procedure as described by Zhu et al. (2010; 2011). We indicated FAZ localization between pedicel and spur (Line 447), only to be made aware for the general audience. However, we tried to avoid taking spur tissues during our sample collection. Again, we agree that sampling was done by hand in the field, and there might be a variation in tissue collection from each fruit sample. In addition, the RNA was extracted from multiple samples in both control and treated fruits from both orchards to increase the RNA content in the final samples. The corrected figure and texts were added in Line 447 to 456.
- Timing of sampling relative to hexanal application: Similarly, to point one above, I think it is worth putting into context in the discussion that the DE analysis described here is a single point intime relative to the hexanal application. This snapshot in time does not really say much about the rate of change of expression from the time of hexanal application (i.e. was the change in expression immediate? Or did it ramp up slowly? Or is what we observed greatly diminished from some maximum). Clearly having more timepoints in the analysis would clear up this uncertainty.
We agree that time course RNA-Seq analysis would give a clearer picture of the relative variation in gene expression. It would certainly provide information on how hexanal message is perceived and decoded by the tissue. However, at this time, our primary objective is to have a single time point analysis, mainly at the commercial maturity stage, to see how hexanal alters the gene expression at the commercial maturity stage. The commercial maturity stage is crucial as the drop significantly increases from previous points (Arseneault and Cline, 2017 and verbal communication with growers). We strongly believe that having baseline information of genes and hormonal changes at this stage would give an overview for further prediction.
- Figure 4: In my opinion Figure 4 “as is” should not be included in the main text of the manuscript. I feel that parts “a” and “c” can easily be explained as text, and part “b” is much too cluttered. The text is cut off in some of the GO names, for example. I think this network “hairball” could be an interesting figure if it can be simplified to perhaps only including the top 20(?) GO categories.
Thank you for your suggestions. The figure 4a and c have been removed and mentioned in the text (Line 197-198; 218; 264-265; 277-278). We have included the interactive plot of the top 30 enriched pathways (FDR, P < 0.05) in figure 4 (Line 208). A hierarchical clustering tree summarizing the correlation among these top 30 significant pathways was included in supplementary figure 1.
- Figure 8: Figure 8 is quite well done, and perhaps would be the jumping off point to say that this really is the proposed model if the FAZ really does have the observed gene DE. With the caveats raised in point 1 above.
We greatly appreciate your comment on figure 8. We strongly believe that the constriction region that we sampled is primarily consisting of the abscission zone cells. At the same time, we acknowledge that a minor portion of pedicel tissue may have been added to the FAZ samples as the sampling was done manually at the field. However, its proportion could be negligible at this point as both control and treated samples were collected in the same way. In addition, we used a strict statistical procedure to detect the DEG between control and treated samples (P < 0.05 and expression fold change should be >2).
- Consistency with software names/versions: There are a few instances of software being mentioned without a reference or version number. In addition, I suggest you mention
Whether default software parameters were used during analysis.
Thank you for notifying these two mistakes. We added the software version and parameters in the appropriate places in the main text (Line 516-525; 527; 532).
- DE normalization and EdgeR: One thing that jumped out at me, having not used EdgeR before, was the lack of details about the read normalization for the DE analysis. I gather this normalization is carried out during its internal modelling, but it is not clear form the text this is the case. A quick mention of normalization would help the reader here since anyone mildly familiar with DE analysis will be looking for something of this nature in the text.
Thank you for your suggestion. We used EdgeR software (v 3.14.0), an empirical analysis of digital gene expression data in R, deposited in DeBrowser (v1.16.1) to detect the DEGs. The normalization of reads counts for individual genes was carried out during the internal modelling by EdgeR package using a weighted trimmed mean of log expression ratios (M values) (TMM) normalization method. This has been mentioned in the materials and method (Line 537-539).
- Maturity of the sampled fruit: There are several mentions about the sampling of the fruit and fruit maturity in the Methods. It is not clear however what criteria was used to assume fruit maturity. Starch/Sugar test?, Fruit colour?
Since we carried out this experiment in two commercial orchards in the Niagara region of Ontario, we followed the maturity index protocol by the grower. They pick the fruits at commercial maturity based on when the ground colour begins to change from green to yellow, and by that time starch index should be at least 6 (Cornell chart index). We have added it in the materials and method section. (Line 459-460)
- Fruit retention and fruit weight: It is not clear from the text whether any measurements were made to see if avg. fruit weight (of dropped fruit) of the test trees was different, leading to confounding of results of the fruit retention analysis.
Unfortunately, we did not take into consideration the relationship between fruit retention and fruit weight (of the dropped fruits) due to many factors. The majority of the dropped fruits were either removed for various purposes or partially consumed by animals. However, the weight all of the fruits used in this experiment was measured. Overall, in our future experimental trials, it would be a good idea to try to connect fruit retention and fruit weight. This suggestion would help us direct our future experiments to correlate the drop fruits and fruit retention.
Round 2
Reviewer 1 Report
The ethylene related data are important for this manuscript, which tries to address hormonal crosstalk.
If authors think that the statistically significant difference of the ethylene level could not be shown due to the methodology used, authors should improve the method or use a suitable method. When p> 0.05, although p< 0.1, the difference is typically not considered as statistically significant. Such data may not up to the publication standard of IJMS. When the data did not show a statistically significant difference, it is not appropriate to have the descriptions “ Hexanal treated fruits produced lower ethylene (21%) than the control fruits “ (line 144) and “Hexanal treated fruits produced less ethylene” (line 340). Instead of suggesting “Hexanal treated fruits produced less ethylene” (line 340) without being supported by the data, authors should consider discussing the ethylene issue solely based on the expression data, which is more solid than the ethylene level measurement data. I would suggest authors to also analyze the expression level of ethylene-inducible genes, which should reflect the ethylene level and show that the ethylene level did exert effect on the system.
The result part is “part 2” while the materials and method is “part 4”, it is confusing that part 2 starts with Figure 2 while Figure 1 is in part 4.
Round 3
Reviewer 1 Report
Authors have addressed the suggestions.
In the latest version of the manuscript, change “we could not measure the ethylene in the FAZ” (lines 407-408) to “the difference of ethylene level in FAZ could not be detected between the control and hexanal treated samples” [authors could measure ethylene in FAZ (Table 2), just that the difference could not be shown]
Author Response
Thank you for the comment. We have edited the lines 407-409 as follows: 'The difference of ethylene level in FAZ could not be significantly detected between the control and hexanal treated samples; however, reduced transcripts of key ethylene biosynthesis genes by hexanal in the FAZ may have reduced the ethylene level and thus potentially contributed to reduction of ABA biosynthesis and signalling in the FAZ, compared to control.'